# Research Progress of α-Glucosidase Inhibitors Produced by Microorganisms and Their Applications

**DOI:** 10.3390/foods12183344

**Published:** 2023-09-06

**Authors:** Fei Ren, Nairu Ji, Yunping Zhu

**Affiliations:** 1Beijing Engineering and Technology Research Center of Food Additives, Beijing Technology & Business University, Beijing 100048, China; 2130022048@st.btbu.edu.cn (F.R.); 2250021016@st.btbu.edu.cn (N.J.); 2School of Food and Health, Beijing Technology and Business University, Beijing 100048, China

**Keywords:** diabetes, α-glucosidase inhibitor, 1-Deoxynojirimycin, sustainable, fermented foods

## Abstract

Based on the easy cultivation of microorganisms and their short cycle time, research on α-glucosidase inhibitors (α-GIs) of microbial origin is receiving extensive attention. Raw materials used in food production, such as cereals, dairy products, fruits, and vegetables, contain various bioactive components, like flavonoids, polyphenols, and alkaloids. Fermentation with specific bacterial strains enhances the nutritional value of these raw materials and enables the creation of hypoglycemic products rich in diverse active ingredients. Additionally, conventional food processing often results in significant byproduct generation, causing resource wastage and environmental issues. However, using bacterial strains to ferment these byproducts into α-GIs presents an innovative solution. This review describes the microbial-derived α-GIs that have been identified. Moreover, the production of α-GIs using industrial food raw materials and processing byproducts as a medium in fermentation is summarized. It is worth analyzing the selection of strains and raw materials, the separation and identification of key compounds, and fermentation broth research methods. Notably, the innovative ideas in this field are described as well. This review will provide theoretical guidance for the development of microbial-derived hypoglycemic foods.

## 1. Introduction

Diabetes is one of the major chronic diseases threatening human health. Approximately 10.5% of adults (20–79 years old) worldwide suffer from diabetes as of the end of 2021, according to the 2021 International Diabetes Federation Global Diabetes Map (10th edition). The study predicts that the number of people with diabetes will exceed 1.31 billion, and the global prevalence of type 2 diabetes is expected to rise by 61.2% by 2050 [1]. Thus, the effective prevention and treatment of diabetes is a top priority. To date, some studies have confirmed that long-term postprandial hyperglycemia is one of the important factors inducing type 2 diabetes [2,3], and it is common to use α-glucosidase inhibitors (α-GIs) for the prevention and treatment of type 2 diabetes. It has been reported that oral α-GIs can delay starch digestion and improve hyperglycemia and diabetic complications [4].

The carbohydrate macromolecules present in food are initially hydrolyzed by α-amylase, found in both saliva and the pancreas, into smaller oligosaccharides or disaccharides with random D-glycosidic bonds. These molecules are then further degraded by α-glucosidase at the brush border of cells located in the small intestine to produce glucose, leading to an increase in blood sugar levels. Taking advantage of this feature, α-GIs can combine with α-glucosidase in a competitive or noncompetitive manner to prevent it from hydrolyzing oligosaccharides and disaccharides and delay blood glucose rise [5,6,7,8]. To date, studies on natural α-GIs mainly focus on plants, and there are more than 5000 articles in the Web of Science database alone, such as alkaloids contained in mulberry leaves [9], soy protein [10], *Tribulus terrestris* extracts [11], and flavonoids extracted from citrus fruits [12]. However, some drawbacks of plant-derived α-GIs are that the content of α-GIs from plants is low, the growth cycle is long, and they may be affected by the environment [13]. During microbial research, it has been observed by several investigators that the fermentation broth displays α-glucosidase inhibitory activity. Moreover, the production cycle of α-GIs by microorganisms is shorter, which is more conducive to industrial production.

In recent years, microbial fermentation to produce α-GIs has been increasingly studied, especially in the use of food-based raw materials as well as food processing byproducts as raw materials for the fermentation production of α-GIs, and the high-value utilization of the byproducts is the key to promoting the sustainability of the resources. In addition, a more scientific understanding of microbial-derived α-glucosidase inhibitors will promote the development of this field, and therefore, it is necessary to conduct a systematic review to provide more references for subsequent research on microbial-derived α-GIs.

Currently, the microorganisms known to produce α-GIs are mainly bacteria and fungi, such as *Actinoplanes*, *Streptomyces*, *Bacillus*, *Penicillium*, and *Aspergillus*. In this review, the identified α-GIs and their inhibitory potentials are summarized. Second, the production of α-GIs by fermentation is presented comprehensively, including an introduction to the production of traditional culture media and fermentation using food-related materials. Third, we present the identification and detection of components and functions from microbial production. In the end, an expansion of other biological activities of microbial-derived α-GIs is concluded. This review will indicate that the use of microbial α-GIs production shows great industrial potential.

## 2. Source of α-GIs

Microbially derived α-GIs can be divided into two main categories: bacterial and fungal. Microbial-derived α-GIs that have been identified are shown in Table 1. 

### 2.1. Bacterially Derived α-GIs

Bacterially derived α-GIs are mainly from *Bacillus* and *Streptomycetes*, such as acarbose, miglitol, and voglibose, which have been used in clinical practice. Some potential compounds have been identified, including 1-deoxynojirimycin (DNJ), homogentisic acid, and genistein. The structures of bacterially derived α-GIs are shown in Figure 1, and we found that most of these compounds are polyhydroxy structures.

**Table 1 foods-12-03344-t001:** Microbial-derived α-GIs that have been identified.

Classification	Name/Type of Compound	Structures	Source	Enzyme Source	IC_50_	References
Bacteria	Acarbose	Figure 1	*Actinoplanes Streptomyces*	Rat α-glucosidases	1.39 ± 0.23 mg/mL	[14,15,16]
	Miglitol		*Streptomyces*	Not displayed	2.43 µM	[17,18]
	Voglibose		*Streptomyces*	Baker’s yeast	23.4 ± 0.3 μM	[19,20]
	1-Deoxynojirimycin		*Bacillus Streptomyces*	Not displayed	0.35 mg/mL	[21,22]
	Homogentisic acid		*Paenibacillus*	Rat α-glucosidases	220 μg/mL	[23]
	Genistein		*Streptomyces*	Baker’s yeast	50 nM	[24]
	CKD-711 and CKD-711a		*Streptomyces*	Rat intestinal mucosa	2.5 μg/mL and 6.5 μg/mL	[25]
	Hemi-pyocyanin		*Pseudomonas aeruginosa*	Yeast	0.572 mg/mL	[26]
Fungus	Polyhydroxy phenolic	Figure 2 (1)	*Aspergillus insulicola*	Not displayed	17.04 ± 0.28 to 292.47 ± 5.87 μM	[27]
	Alkaloids/meroterpenoids/steroids	Not displayed	*Aspergillus*	Saccharomyces cerevisiae recombinant	0.3 ± 0.0.03 to 0.4 ± 0.02 mM	[28]
	Prenylated aspulvinones	Figure 2 (2)	*Aspergillus terreus*	Saccharomyces cerevisiae	2.2 and 4.6 μM	[29]
	Polyketides	Figure 2 (3)	*Preussiaminimoides*	Yeast	2.9 to 155 μM	[30]
	Diphenyl ethers and phenolic bisabolene sesquiterpenoids	Figure 2 (4)	*Aspergillus flavus*	Not displayed	1.5–4.5 M	[31]
	Polyketides and phenylpropanoid	Figure 2 (5)	*Pseudolophiostoma*	Not displayed	48.7–120 µM	[32]
	Xanthones	Figure 2 (6)	*Penicillium canescens*	Saccharomyces cerevisiae type I	32.32 ± 1.01 to 75.20 ± 1.02 μM	[33]
	Phenolic	Not displayed	*Alternaria destruens*	Not displayed	Not Displayed	[34]
	Aromatic compounds	Figure 2 (7)	*Hericium erinaceus*	Saccharomyces cerevisiae	<20 μM (essential compound)	[35]
	Polysaccharides	Not Displayed	*Pleurotus eryngii*	Not displayed	Not displayed	[36]

#### 2.1.1. Structural Analogs of Glucose

α-GIs of bacterial origin are mainly structural analogs of glucose, including acarbose, miglitol, voglibose, and DNJ.

Acarbose is a pseudosaccharide (C_25_H_43_NO_18_) produced by microbial fermentation that can competitively bind to the catalytic site of α-glucosidase and inhibit the decomposition of starch, polysaccharides, etc. In addition, acarbose can lower glycemia and glycated hemoglobin [15,37] and significantly increase the reversion of impaired glucose tolerance to normal glucose tolerance (*p* < 0.0001) [38]. In 1977, acarbose was extracted from the metabolites of the *Actinomycete acetinoplanesst* strains SE50, SE18, and SE82. This compound was further developed as a hypoglycemic drug by Bayer in Germany. In 1995, it was approved by the Food and Drug Administration. It became the first α-GI to be applied to clinical treatment. However, during clinical treatment, acarbose exhibited some adverse reactions. Because acarbose inhibits the activity of α-glucosidase in the small intestine, some undigested saccharides in the intestine are fermented under the action of intestinal microorganisms, and adverse reactions, such as abdominal distension, diarrhea, and constipation, may occur after medication [39]. On the one hand, there is currently the utilization of natural products in combination with acarbose to attenuate its side effects [40], while on the other hand, the search for safer and more promising α-GIs is imminent.

In 1982, miglitol, a derivative of DNJ, was extracted from the fermentation broth of *Streptomyces lavendulae*. Its mechanism of action is a competitive inhibitor of α-glucosidase in small intestinal epithelial cells, and it is currently used as a second-generation drug for the treatment of type 2 diabetes [41]. Voglibose is the main component of the glucose-lowering drug Besin tablets independently developed by Takeda Pharmaceutical Industry Co., Tokyo, Japan. It is an amino sugar analog isolated from the fermentation broth of *Actinomyces*, and it is also a new and efficient α-GI [16]. Valigobose may also facilitate mobile α endogenous glycogen-like peptide 1 (GLP-1), which has an inhibitory action on glycogen, thus lowering fasting glucose levels. GLP-1 is also an insulinotropic hormone that enhances insulin secretion and insulin sensitivity [42]. Miglitol and valigobose face the same dilemma as acarbose, which brings challenges to the clinical dosage.

DNJ, the fractional formula of which is C_6_H_13_NO_4_, is a potent α-GI and a key active substance with hypoglycemic function in mulberry leaves. As early as 1976, Yagi et al. [43] isolated DNJ from the root bark of mulberry. To date, scientific researchers have carried out detailed research on the special effects of DNJ on hypoglycemic function, and its safety has also been verified in human experiments [21,44]. To date, there are four ways to obtain DNJ: (i) plant sources, such as mulberry leaves [45]; (ii) animal sources, such as sericulture [21], which enrich and accumulate DNJ in the body through eating mulberry leaves; (iii) chemical synthesis [46]; and (iv) microbial sources, which are more suitable for large-scale production of DNJ due to their short growth cycle, easy cultivation, and relatively low cost. DNJ has been identified from *Bacillus*, *Streptomyces*, and *Actinomyces*. In addition, DNJ still has a long way to go from theory to large-scale application. On the one hand, the challenge of obtaining high-purity DNJ samples from microbial fermentation broth through more efficient separation and purification methods remains to be addressed. The existing literature primarily focuses on the preparation of crude DNJ samples, with limited studies on the production of high-purity DNJ samples. On the other hand, there is a need to expand the selection of DNJ-producing bacteria and explore fermentation optimization strategies to further enhance their productivity.

#### 2.1.2. Other α-GIs

Several researchers in the field have successfully purified and identified additional inhibitors using various methods, including column chromatography. For instance, novel α-GIs—CKD-711 and CKD-711a—were purified using Dowex 50W-2X and Sephadex G-10 chromatography, exhibiting IC_50_ values of 2.5 μg/mL and 6.5 μg/mL, respectively [25]. In a study by Lee et al. [24], genistein was isolated from secondary metabolites of *Streptomyces*, demonstrating noncompetitive enzyme inhibition of α-glucosidase. Nguyen et al. [23] employed *Paenibacillus* sp. TKU042 for fermentation with 0.1% squid pens as the sole carbon/nitrogen source. They isolated a major inhibitor from fermented squid pen powder and identified it as homogentisic acid (HGA). HGA was found to be a nonsugar-based α-glucosidase inhibitor (IC_50_ = 220 μg/mL) and exhibited stronger activity than acarbose.

### 2.2. Fungus-Based α-GIs

To date, α-GIs have been identified from fungi, including Aspergillus, Penicillium, Alternaria, Pseudolophiosptoma, and some mushrooms. Figure 2 shows the structures of representative compounds.

**Figure 2 foods-12-03344-f002:**
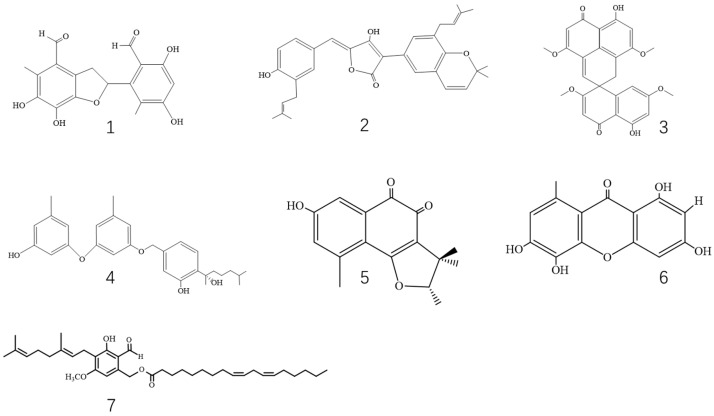
Structures of fungus-based α-GIs possessing α-glucosidase inhibitory activity (only the highly promising α-glucosidase inhibitors identified are shown).

In their study, Zhao et al. [27] conducted fermentation using *Aspergillus* and discovered that various polyhydroxyphenolic compounds isolated from the fermentation cultures exhibited α-glucosidase inhibitory activity. The IC_50_ values for these compounds ranged from 17.04 ± 0.28 to 292.47 ± 5.87 µM. Similarly, Shan et al. [28] isolated an active substance with stronger α-glucosidase inhibitory activity (IC_50_ values ranging from 0.3 ± 0.03 to 0.66 ± 0.04 mM) than the positive drug acarbose from the fermentation broth of *Aspergillus terreus*. Wu et al. [31] isolated two new diphenyl ethers and four new phenolic bisabolane sesquiterpenoids (4) from the culture of *Aspergillus flavus* QQSG-3, in this study, with IC_50_ values ranging from 1.5 to 4.5 μM. Wu et al. [29] isolated three new prenylated aspulvinones V–X from the fermentation broth of *Aspergillus terreus* ASM-1. The IC_50_ values of the two most promising α-GIs were determined to be 2.2 and 4.6 μM.

In addition, Allan et al. [32] investigated α-glucosidase inhibitory activity in ethyl acetate extracts of a *Pseudolophiosptoma* sp. and isolated three phenalenones and one phenylpropanoid. Among them, scleroderolide (5) showed the most potent α-glucosidase inhibitory activity, with an IC_50_ value of 48.7 μM. Manuel et al. [30] isolated two new polyketides with novel structures, minimoidiones A and B, along with the known compounds preussochromone C, corymbiferone, and 5-hydroxy-2, 7-dimethoxy-8-methylnaphthoquinone, from the grain-based culture of the endophytic fungus *Preussia minimoides*. Compounds 1–4 demonstrated significant inhibition of yeast α-glucosidase, with IC_50_ values ranging from 2.9 to 155 μM.

Furthermore, Abd. et al. [33] reported one previously unreported xanthone and two known xanthones as α-GIs, isolated from an endophytic *Penicillium canescens* obtained from fruits of *Juniperus polycarpos*. The IC_50_ values for these compounds ranged from 32.32 ± 1.01 to 75.20 ± 1.02 mΜ. Jasleen et al. [34] isolated two fractions (AF1 and AF2) from an endophytic fungus *Alternaria destruens*, which possessed α-glucosidase inhibitory activity.

Currently, the hypoglycemic effect of mushrooms has been reported, and researchers have identified a variety of α-GIs from mushrooms [47,48]. Seul et al. [35] isolated ten compounds from the fruiting bodies of *H. erinaceus* and found that erinacenol D, 4-[3′,7′–dimethyl-2′,6′-octadienyl]-2-formyl-3-hydroxy-5-methyoxybenzylalcohol, hericene A, hericene D, and hericenone D strongly inhibited α-glucosidase activity, with IC_50_ values <20 μM. Zheng et al. [36] isolated polysaccharides (PEBP-II and PEMP-II) from the fruiting body and mycelia of *Pleurotus eryngii* and demonstrated α-glucosidase inhibition levels of 56.13% and 57.57%. When the polysaccharide concentration was 6 mg/mL, the results indicated that they could be utilized as potential inhibitors of α-glucosidase.

These studies highlight various compounds and sources that have been identified and investigated for their α-glucosidase inhibitory activity.

## 3. Screening and Evaluation of the Potential Strains

Firstly, it is critical to consider the biological characteristics, fermentation potential, and product acceptability of the strains. The biological characteristics encompass factors such as coagulation ability, acid and alkali resistance, bile salt tolerance, and self-aggregation capacity compared to pathogenic bacteria. For example, a study investigated the impact of several *Lactobacillus* and *Bifidobacteria* on whey beverage fermentation, revealing that *Lactobacillus acidophilus* La-5 and *Bifidobacterium bifidum* Bb-12 exhibited higher α-glucosidase inhibitory activity, improved the high-saturation high-cholesterol index, and produced peptides with ACE inhibitory, antimicrobial, immune-modulatory, and antioxidant activities [49]. The fermentation potential of strains can be evaluated by simulating the gastrointestinal environment to determine their survival rate and α-glucosidase inhibitory activity. Chandana et al. isolated nine strains from fermented sugarcane juice and found that the survival rate of intestinal strains was above 98% in simulated gastric acid, with α-glucosidase inhibition ranging from 21% to 85% [50]. Furthermore, certain researchers have conducted oral administration of *Bacillus coagulans*, a probiotic, directly to mice [51]. The presence of α-glucosidase inhibitory activity in the intestinal contents and feces of the mice confirmed the colonization of the intestine by this strain and its ability to produce α-GIs. Sensory acceptance is also an important assessment parameter, which can be evaluated through sensory evaluation or sensory analysis to assess the volatile components of fermented products [52].

Secondly, assessing the safety of the selected strains is crucial. This can be achieved through methods such as DNA/DNA hybridization and 16S RNA sequencing for accurate strain identification. Additionally, evaluating antibiotic sensitivity, adherence to host tissues, and hemolysis assays, along with conducting animal experimental models, can help determine safety. These methods are commonly used to assess strain safety [53,54]. Moreover, some researchers have assessed strain safety at the genetic level. For example, Nipa and Dina employed databases like KEGG to search for and identify virulence factors, undesirable genes, resistance genes, transfer potential, and antimicrobial product production capabilities, providing a comprehensive procedure for analyzing strain safety at the genetic level [55,56]. In the future, integrating in vitro and in vivo approaches and leveraging emerging technologies, such as transcriptomics, genomics, and metabolomics, would enable a comprehensive evaluation of strain safety, further ensuring the safe utilization of these strains.

## 4. Fermentation Production and Potential Applications of α-GIs

Microbial production has the advantages of a short life cycle, low development cost, and easy access to abundant active ingredients. The development of highly efficient, highly safe, and high-yield α-GIs from microbial products has broad market prospects with the continuous innovation and development of biotechnology and separation processes [57,58,59].

### 4.1. Fermentation Using Traditional Culture Medium

In this section, the production optimization strategies of acarbose and DNJ that have been widely studied are summarized to provide a reference for the development of new α-GIs.

#### 4.1.1. Production of Acarbose

The culture conditions of acarbose are shown in Table 2. Maltose is the key carbon source for acarbose synthesis. For example, Byoung et al. [60] maintained the fermentation broth maltose at a high concentration level to effectively produce acarbose and found that the maximum value obtained at an osmotic pressure of 500 mOsm/kg was 3200 mg/L. Wang et al. [61] found that maltose, glycerol, and monosodium glutamate favored acarbose production, yielding 4210 mg of acarbose at 591 mOsm/kg osmolarity. In addition, a fed-batch culture process under preferential osmolality was constructed through intermittently feeding broths with feed medium consisting of 14.0 g/L maltose, 6.0 g/L glucose, and 9.0 g/L soybean meal at 48 h, 72 h, 96 h, and 120 h, and the yield of acarbose reached 4878 mg/L [62]. Li et al. [63] increased the final yield of acarbose to approximately 5000 mg/L by fermentation for 168 h at a high total sugar concentration (75–80 g/L), pH 7.0–7.2, and DO concentration of 40–50%. In conclusion, the application of strategies such as batch replenishment, optimization of osmolality, and maltose concentration prove advantageous for enhancing the production of acarbose.

#### 4.1.2. Production of DNJ

We summarize the current research on DNJ-producing microorganisms and yield enhancement optimization in Table 3. It can be seen from the table that optimization by screening nitrogen and carbon sources, cultivation conditions, and other factors and using single factor and response surface optimization is the more classical method to increase yield. In addition, there are reports on the overexpression of key genes that control DNJ synthesis, mutation breeding, metabolic engineering, and other gene levels to increase DNJ production. For example, Nguyen et al. [64] isolated a mutant *Bacillus subtilis* strain I.247 with a 52% increase in DNJ production after ultraviolet random mutagenesis. Based on a single factor and response surface optimization, the DNJ concentration in the optimized medium containing 3.4% sorbitol and 2.4% yeast extract as carbon and nitrogen sources reached 359 mg/L after 5 days of cultivation at 32 °C. Finally, *Bacillus subtilis* I.247 transformed with the vector carrying the gabT1-yktc1-gutB1 gene could produce 773 mg/L DNJ under the same culture conditions. Furthermore, a recent study has introduced a pioneering approach to predicting the DNJ synthesis pathway in strains by employing isotope labeling and nuclear magnetic resonance techniques. This innovative method aims to identify the precursors associated with DNJ synthesis, and subsequently enhance DNJ production by incorporating metabolic pathway inhibitors at specific time points. The results obtained by the researchers demonstrate a significant improvement in DNJ production [65]. This optimization strategy holds promising potential for its application in other strains, thus enabling the achievement of high levels of DNJ production in the future.

### 4.2. Fermentation Using Food-Related Materials

Considering the fermentation cost and the production of functional food containing α-GIs, various food-related materials are used as the growth medium for the microorganisms. To date, in addition to acarbose and DNJ, numerous researchers have found other potential α-GIs when developing functional foods by fermentation. Figure 3 shows the schematic diagram of fermentation for producing α-GIs using food-based raw materials and byproducts. They use inexpensive noncompetitive carbon sources or industrial raw materials with low utilization rates for fermentation and obtain different degrees of excellent results. The relevant research in recent years is summarized as follows. Table 4 shows the strains that can produce α-GIs through metabolism, including *Bacillus*, *Streptomyces*, *Lactobacillus*, and other common bacteria, and fungi, such as yeast and mushroom mycelium. The sources of fermentation materials are diverse, including fruits, grains, and dairy products that are common in our daily lives. There are also some processing byproducts and aquatic products that are used in fermentation.

#### 4.2.1. Fermentation Using Grains

Grain fermentation products have a long history. A group of researchers explored the effect of dietary fiber intake and diabetes incidence over 13.8 years and concluded that higher cereal fiber intake was associated with a 25% lower incidence of type 2 diabetes [73]. In addition, resistant starch, protein, and some phenolic compounds contained in grains are also the main antidiabetic substances [74]. To date, there are relatively many related studies on the usage of grain fermentation to obtain α-GIs. Most current studies on the production of α-GIs through the fermentation of cereals focus on single-strain fermentation. This process yields various α-GIs, including bioactive peptides [75], polyphenols [76], isoflavonoids [77], and alkaloids [78]. These active compounds are derived from both the food ingredients themselves and the secondary metabolites of the microorganisms. For instance, it has been found that a majority of the phenolic compounds present in chickpea seeds are in a bound form, and the fermentation strategy promotes the release of these phenolic compounds [79]. Others have utilized the *Aspergillus oryzae* and *Aspergillus niger* for solid-state fermentation of lentils. The lentil extracts produced by fermentation with *A. niger* after 48 h were able to inhibit the α-glucosidase activity by up to 90%, and it was found that the changes in total phenol content of lentil extracts correlated well with the increase in biological activity, which may be because enzymes produced by *Aspergillus* metabolism can effectively hydrolyze the plant cell wall matrix, which promotes the release of polyphenols and other actives [80]. Obaroakpo et al. investigated the α-glucosidase inhibitory activity of germinated protein hydrolysates of quinoa yogurt beverage, and among the isolated and identified peptide activity fractions (Table 4), QLCSY13 exhibited the highest α-glucosidase inhibitory activity (IC_50_ = 8.86 mg/mL) [81]. Moreover, the application of multi-strain co-fermentation has demonstrated a positive impact on the production of α-GIs. For instance, researchers have explored the co-fermentation of quinoa seeds utilizing combinations of Limosilactobacillus and Lacticaseibacillus rhamnosus strains. The resulting α-glucosidase inhibitory activity (62.3%) surpassed that achieved through single-strain fermentation (50.7% and 56.8%) [82].

**Table 4 foods-12-03344-t004:** Development and utilization of fermented foods rich in α-glucosidase inhibitory activity.

Category	Raw Material	Strain	Type of Extract	Major Active Substance	IC_50_/Inhibition Ratio	Yield	References
Cereal	Rice bran	*Bacillus subtilis* MK15	Separation and purification using ultrafiltration, DEAE Sepharose, and Sephadex G-25	Oligopeptide	1 mg/mL was 15%	Not mentioned	[75]
	Black soya bean	*B. subtilis*	23% ethanol and 2% formic acid extract	Phenolic compound	0.353 mg GAE/mL	about 25 mg GAE/g	[76]
	Soybean	*H. erinaceus* and *H. ramosum* wild mushroom fruiting bodies	Ethanol extract	Genistein	62.3–79.5%	1.465 ± 0.033 μg/g	[77]
	Okara	*B. amyloliquefaciens* SY07	Crude extract	1-Deoxynojirimycin	0.454 mg/mL	Not mentioned	[78]
	Chickpea	*Lactiplantibacillus plantarum*	Fermentation supernatant	Phenolic compound	Probably less than 55%	0.7-1.03 mg GAE/g	[79]
	Lentils	*Aspergillus oryzae* and *Aspergillus niger*	Distilled water extract	Phenolic compound	90%	4.27 mg GAE/g	[80]
	Quinoa seeds	*Lactobacillus casei* SY13	Germinated protein hydrolysates of quinoa yogurt beverage	Peptide	8.86 mg/mL	Not mentioned	[81]
	Quinoa seeds	*Limosilactobacillus* and *Lacticaseibacillus*	Ethanol extract	Not mentioned	62.30%	Not mentioned	[82]
	Okara	*Bacillus subtilis B2*	*Separation and purification by multi-column chromatography*	1-Deoxynojirimycin	0.2 mg/ml	Not mentioned	[83]
	Soybean	*B. amyloliquefaciens* HZ-12	Methanol extract	1-Deoxynojirimycin	Not mentioned	870 mg/kg	[84]
	Tartary buckwheat	*L. plantarum* and *L. paracasei*	70% ethanol extract	Phenols and flavonoids	0.51 mg/mL	251.8 ± 10.4 mg/g and 25.7 ± 0.4 mg/g	[85]
Garden stuff	Extract of capsicum	Not mentioned	80% ethanol extract, and purified with AB-8 macroporous resin	Phenolic compounds	0.83 mg/mL	546.70 ± 20.53 (GAE)/100 g	[86]
	Pumpkin	*Lactobacillus mali* K8	Fermentation supernatant	Not mentioned	95.89 ± 0.30%	Not mentioned	[87]
Garden stuff	Pepper	*Bacillus licheniformis* SK1230	Water extract	Phenolics	14.47 ± 0.67%	Not mentioned	[88]
	Fruit enzyme	Natural fermentation	Fermentation supernatant	Not mentioned	81.67%	Not mentioned	[89]
	Apple juice	*Lactobacillus*	Not mentioned	Not mentioned	21.51%	Not mentioned	[90]
	Yacon	*Lactiplantibacillus plantarum* NCU001043	Fermentation supernatant	Phenolic compounds	740 μg AE/mL	9.7 mg GAE/mL	[91]
Aquatic products	Chitin as a carbon source	*Paenibacillus* sp. TKU042	Fermentation supernatant	Not mentioned	81 μg/mL 93%	Not mentioned	[92]
	Squid ring	*Paenibacillus*	Separation and purification using Diaoin, Octadecylsilane opened columns, and preparative HPLC	Homogentisic acid	220 μg/mL	Not mentioned	[23]
	Laminaria	*Saccharomyces cerevisiae* and *Lactiplantibacillus*	Crude extract	Not mentioned	0.58 ± 0.018 mg/mL	Not mentioned	[93]
Other	Rubing cheese	*Lactobacillus*	Crude extract	Polypeptide	0.821 mg/mL	Not mentioned	[94]
	Artemisia capillaris	*Leu. mesenteroide*	Fermentation supernatant	Not mentioned	76.05%	Not mentioned	[95]
	Camel and beef sausages	*L. lactis* KX881782	Water-soluble extract	Not mentioned	About 40%	Not mentioned	[96]
	Soybean leaves	*Lactobacillus plantarum* P1201 and *Lactobacillus brevis* BMK184	Crude extract	Total phenolic, total flavonoid, and isoflavone aglycone	49.86%	Not mentioned	[97]
	Cassava wastewater	*Serratia marcescens* TNU01	Separation and purification using ethyl acetate for layer separation and silica open column	Prodigiosin	0.0183 μg/mL	6150 mg/L	[98]
	Hibiscus sabdariffa	*Saccharomyces cerevisiae*	Crude extract	Delphinidin 3-*O*-sambubioside	543 μmol/L	Not mentioned	[99]

Soybean fermented foods have been extensively studied due to their rich nutritional composition, including soybean protein, dietary fiber, isoflavones, oligosaccharides, and other beneficial nutrients. Soybean protein has been found to have the ability to promote insulin secretion, while soy isoflavones have shown potential in improving insulin sensitivity [100]. As a result of these properties, soybean fermented foods, such as soybean paste, tempeh, sufu, and natto, exhibit hypoglycemic functional properties [101]. In the process of fermentation, it has been discovered that some of the main α-GIs may originate from secondary metabolites produced by microorganisms. For example, certain bacteria that produce DNJ utilize soybeans as a solid-state fermentation medium, leading to high yields of DNJ. The presence of these α-GIs further contributes to the hypoglycemic effects of fermented soybean products [84]. Chen et al. [102] prepared tempeh in the laboratory with three fungal strains, *Aspergillus oryzae*, *Actinomyces cavitalis*, and *Rhizobium rhizogenes*, and found that the α-glucosidase inhibitory activity of all soybeans increased slightly and that the inhibitory activity of homemade tempeh on α-glucosidase partly depended on the fungal strain and the salt used for fermentation. On the other hand, soybean residues are produced during soybean processing. Although people have been aware of the potential benefits of these byproducts, the current utilization of these byproducts is still insufficient [103]. Soybean residue contains proteins, lipids, carbohydrates, and bioactive compounds, which provide high-quality nutrients for the growth and metabolism of strains [104]. For example, Zhu et al. [83] first isolated DNJ from soybean residue fermented by *Bacillus subtilis*.

Through extensive research on grain fermentation, it becomes evident that the utilization of strain fermentation can effectively facilitate the generation and release of compounds possessing α-glucosidase inhibitory activity. This phenomenon contributes to the enhancement in the overall nutritional value of cereals. Simultaneously, the grains provide essential nutrients required for the growth of the fermenting strains, leading to the production of diverse secondary metabolites endowed with α-glucosidase inhibitory properties. Nevertheless, it is important to acknowledge that the exploration of fermentation products harnessing α-glucosidase inhibitory substances is still limited. While certain studies have identified the presence of potential α-GIs, such as polyphenols, genistein, and hydroxydaidzein [77,85], the exact nature of the key active compounds remains undisclosed. This highlights the need for future investigations to delve deeper into this area of research in order to identify and elucidate the pivotal constituents responsible for the observed α-glucosidase inhibitory effect.

#### 4.2.2. Fermentation Using Fruits and Vegetables

Fruits and vegetables are indispensable food sources for daily life, and there is evidence that several fruits and vegetables can inhibit the activity of these human digestive enzymes in vitro [6]. Fermented fruits and vegetables are rich in vitamins, polyphenols, and other nutrients [105,106]. To date, we are familiar with fruit wine, enzymes, pickles, etc., which are fermented fruits and vegetables. After fermentation by *lactobacillus*, fruits and vegetables can improve their flavor and nutritional value [88].

In the field of fruit and vegetable fermentation, probiotic fermentation is the focus of research. Therefore, researchers have primarily studied the α-glucosidase inhibition potential, sensory perception, and strain survival rate of fermented products after simulating the intestinal environment [89,90]. For instance, a study used pumpkin as a raw material and fermented it with *Lactobacillus mali* K8. The resulting high-quality drink exhibited 95% inhibition activity against α-glucosidase and an 88% survival rate of *L. mali* K8 after 4 weeks of cold storage while also receiving positive sensory acceptance [87]. Moreover, various methods have been employed to explore the fermentation products that possess α-glucosidase inhibitory properties. For example, Peng et al. fermented snow lotus using *Lactiplantibacillus plantarum* NCU001043, resulting in the production of novel phenolic compounds. Spearman’s correlation analysis suggested that 4-hexyloxyphenol could be the key α-GI [91]. Finally, molecular docking has been utilized to investigate the mechanisms of action of these α-GIs. In one study, Li et al. examined the bioactivity changes during the production of pickled chili peppers. Their findings indicated that phenolic compounds, particularly luteolin, exhibited the strongest inhibitory effect on α-glucosidase, with an IC_50_ of 49.31 ± 8.86 μmol/L. Molecular docking analysis revealed that this inhibition might be attributed to the formation of hydrogen bonds and hydrophobic interactions between these compounds and the key amino acid residues of α-glucosidase [86].

In addition, many byproducts are produced during the processing of fruits and vegetables, especially in the winemaking process. For example, the wine industry associated with the Mediterranean diet is reported to produce more than 9 million tons of grape pomes per year, which is an environmental challenge. Grape pomace is a potential source of α-GIs because it is rich in polyphenols and fiber [107,108]. Based on this feature, researchers have developed milk kefir fortified with Sangiovese red grape residue [109]. Not only in the wine industry but also in other fruit and vegetable processing, different degrees of processing byproducts are produced, and these byproducts often contain a variety of biological activities [110]. In the future, this may be a great way to use these processing byproducts by strain fermentation for high-value utilization.

#### 4.2.3. Fermentation Using Aquatic Products

Marine resources are a huge treasure trove for human beings. There are a variety of hypoglycemic active substances in marine food, such as algae, which are widely studied [111]. Seaweed bioactives, including polysaccharides, pigments, fatty acids, polyphenols, and peptides, have been proven to possess various beneficial biological properties that could potentially contribute to functional food and nutraceutical development [112]. At the same time, some studies have shown that the use of strain fermentation can improve its biological functional activity. For example, Yue et al. [93] conducted a two-step fermentation of kelp by *Saccharomyces cerevisiae* and *Lactobacillus* sp., and the inhibition activity of α-glucosidase was improved after fermentation. The upregulation of protocatechuic acid, gallic acid, and isoerucic acid levels explains the increased inhibitory activity of α-amylase and α-glucosidase.

The same is true for the nutritional value of marine animals. With the improvement in the quality of human life, the demand for aquatic products has increased, but this has come with great environmental pressure. It has been reported that the amount of shrimp and crab waste has increased sharply after processing. Squid processing also produces numerous byproducts, which account for 35% of the total catch mass, including head, viscera, skin, and bones. The processing and utilization of these byproducts are also the focus of research [113], and strain fermentation in seafood is still an effective way to solve this problem. In a study conducted by Nguyen et al., fish processing byproducts, such as chitin-containing materials, squid circles, and shrimp head powder, were fermented using *Paenibacillus* sp. TKU042 [23,92,114]. This fermentation process increased the bioavailability of these byproducts and led to the production of α-GIs.

#### 4.2.4. Fermentation Using Dairy Products

Research in the field of dairy α-GIs primarily focuses on bioactive peptides. Peptic digestion of whey proteins can generate peptides with inhibitory properties against α-glucosidase activities [115]. Zhao et al. [116] isolated new α-glucosidase inhibitory peptides from buffalo casein hydrolyzed by Dregea sinensis protease using reversed-phase high-performance liquid chromatography (RP-HPLC) in their study. Through molecular docking, they found that four peptides may inhibit α-glucosidase activity by occupying its potential catalytic active site through the formation of hydrogen bonds and hydrophobic interactions.

Currently, there are limited studies on α-GIs in fermented dairy products. One of the studied products is kefir, a fermented beverage derived from the fermentation of cow, goat, sheep, camel, or buffalo milk using a microbial consortium of *Lactobacillus* and Yeast. Kefir has been found to have antidiabetic effects, leading to a reduction in blood glucose levels, fasting blood glucose, and glycosylated hemoglobin. It can serve as an adjuvant treatment for diabetes [117]. For instance, Wee et al. [118] isolated eight strains of *Lactobacillus* from water kefir grains, and among them, *L. mali* K8 exhibited potent α-glucosidase inhibition potential (cell-free supernatant showed 39.4% glycosidase inhibitory activity).

#### 4.2.5. Fermentation Using Tea

Tea, with its origins in China and a long-standing history of consumption, has been the subject of extensive research exploring its potential health benefits. Numerous studies have indicated that tea consumption can have positive effects on managing hyperglycemia [119]. For instance, Li et al. [120] conducted a study on Cerasus humilis (Bge.) Sok leaf tea and identified myricetin, quercitrin, isoquercitrin, and guajavarin as prominent α-GIs. This finding highlights the potential of tea as a natural source of compounds with therapeutic potential. Moreover, investigations have further focused on the impact of tea fermentation on the production of α-GIs and their potential applications.

Metabolomic studies conducted by Wen et al. [121] during the fermentation process of pickled tea revealed that glycosides and organic acids demonstrated potent inhibitory effects on α-amylase, while catechins and flavonoids effectively inhibited α-glucosidase. Building on this, Wang et al. [122] explored the deep fermentation of dark tea using *Aspergillus nigra*, resulting in the production of high-theabrownin instant dark tea with notable α-glucosidase inhibitory properties. In a separate study, Wang et al. [123] examined the fermentation of guava leaf tea using *Monascus anka* and *Saccharomyces cerevisiae* and observed that enzymatic hydrolysis stimulated the release of phenolic substances, including quercetin and flavonoid aglycones.

In summary, tea fermentation has been identified as an effective approach to enhance the formation of active components with α-glucosidase inhibitory properties. These findings contribute to the exploration of natural α-GIs and their potential application in managing hyperglycemia. However, further investigations are necessary to elucidate the underlying mechanisms and evaluate the clinical implications of these findings.

#### 4.2.6. Fermentation Using Other Materials

In strain fermentation for the production of α-GIs, various other raw materials are also utilized [94,95,96,97,98,99]. Dong et al. [124] developed chestnut for the first time by using *Bacillus natto* to perform solid-state fermentation. Under the optimal conditions, the content of total phenol, total flavonoid, and vitamin C in the fermented product was increased, with high antioxidant activity and increased α-glucosidase inhibitory activity (the IC_50_ reached 53.09 mg/mL). Tran et al. [98] used cassava wastewater as the base medium to forage *Streptomyces* TNU01 to produce pro bacitracin. The pro bacitracin obtained after fermentation showed stronger α-glucosidase inhibition (the IC_50_ reached 0.0183 μg/mL) and better binding energy than acarbose, a commercial anti-diabetic drug.

Clearly, the majority of fermentation studies on food-based and food-processing byproducts have focused on crude products, with limited research on the study of pure compounds. This is an area that requires further advancement in the future. To achieve this, it will be crucial to employ various isolation and purification strategies, such as chromatography and spectroscopic techniques, to separate and identify the structures of specific substances in fermented foods. By conducting such purification processes, researchers can gain a clear understanding of the specific substances responsible for glycosidase inhibition. This knowledge then serves as a foundation for subsequent experiments involving cell culture and animal models, enabling further exploration of the mechanism and potential therapeutic applications of these compounds.

## 5. Identification and Detection of Components and Functions from Microbial Production

### 5.1. Purification and Identification of α-Gis

Fermentation broths are highly complex and contain numerous secondary metabolites, which often pose challenges for purifying desired α-Gis. To overcome this, several general steps for purifying microbial-derived α-Gis have been summarized (Figure 4). Firstly, optimizing the yield of α-GI production is crucial to enhance the enrichment of α-Gis and minimize the production of byproducts. For example, in the production of acarbose, researchers have employed strategies, such as adding inhibitors and controlling osmotic pressure, to increase acarbose production while reducing the production of byproducts like component C [60].

To purify fermented samples, centrifugation, alcohol precipitation, or acid precipitation are commonly used to remove impurities and enrich the desired substances. Researchers have developed specific purification strategies for isolating unknown compounds. For instance, Zhu et al. [83] isolated and purified DNJ from soybean residue fermentation broth by intercepting dialysate containing the desired substances using a molecular range dialysis bag. They then employed techniques such as activated carbon chromatography, CM Sepharose chromatography, and thin-layer chromatography to separate and extract the active compound. Finally, the compound was freeze-dried to obtain the pure product.

Currently, there are various methods available for sample purification, such as semi-preparative liquid phase and column chromatography, which can effectively separate and purify most substances. The purified components are further analyzed using techniques like LC-MS, NMR spectroscopy, and infrared spectroscopy to determine their molecular weights and structures. In a study conducted by Ren et al. [125], secondary metabolites from *Trametes* sp. ZYX-Z-16 were isolated, and their structures were elucidated using high-resolution electrospray ionization mass spectrometry (HRESIMS), 1H nuclear magnetic resonance (1H NMR), and infrared spectroscopy.

### 5.2. Detection and Analysis of α-Gis

To further explore the changes in the content of critical compounds at different stages of microbial fermentation, optimize the fermentation conditions, and adjust the fermentation process, LC-MS/MS is often used [126,127]. For example, LC-MS and metabolomics are often used to study the process of tea heap fermentation. By employing untargeted metabolomics, it becomes possible to evaluate the alterations in fermentation broth composition at various stages. Furthermore, through comparative analysis, it becomes feasible to predict the types of compounds that are linked to α-glucosidase inhibitory activity. An et al. [128] used ultra-performance liquid chromatography–Q Exactive Orbitrap/mass spectrometry to study the changes in chemical components in fermented green tea. Untargeted metabolomics analysis revealed significant changes in the levels of primary secondary metabolites in instant green tea from day 3 to day 5 of fermentation, and targeted metabolomics analysis showed that phenolic acids and free amino acids were positively correlated with α-glucosidase inhibition. Yu et al. [129] studied *Eurotium cristatum* YL-1 solid-state fermented buckwheat and studied the difference in metabolites between nonfermented buckwheat and vine-fermented buckwheat based on nontargeted metabolomics of LC-MS/MS and found that there were significant differences between fermented and unfermented buckwheat. Most phenolic compounds and alkaloids were significantly upregulated during solid-state fermentation.

Currently, LC-MS/MS is the most commonly used means for the analysis of key compounds of fermentation components. In addition, other assays have emerged. For known compounds, such as polyphenols, DNJ, and isoflavone [80,81], the content can be determined using high-performance liquid chromatography (HPLC), while for unknown compounds, some researchers have analyzed the differences in α-glucosidase inhibition activity and HPLC fingerprints before and after fermentation, which allows for the preliminary prediction of information such as the time to peak of key active compounds [23]. Recently, a novel assay was developed to investigate potential components and functions. High-performance thin-layer chromatography (HPTLC) was used to analyze potential functional compounds and active components, and a high-performance thin-layer chromatography–mass spectrometry (HPTLC–MS)-based method was used to characterize unknown bioactive compounds [130]. This method may be useful for future studies of microbial-derived α-gIs.

## 6. Applications of Microbial-Derived α-gIs

Previously, we have presented the utilization of food-based raw materials and processing byproducts for the production of α-gIs with applications in various fields such as hypoglycemic food development and drug discovery. In this section, we will discuss the extended biological activity of microbially derived α-gIs, providing valuable insights for the future development and utilization of these compounds.

In addition to regulating blood glucose, α-gIs isolated from microbial cultures have shown other biological activities. For example, α-glucosidase is involved in the synthesis of glycoproteins, which are key proteins in the viral growth cycle, since the outer coat of some animal viruses is composed of one or two key glycoproteins. The secretion, assembly, and infection of virions are related to glycoproteins, and α-gIs can be used to treat viral diseases [131]. For example, N-linked oligosaccharide processing plays a crucial role in the early stages of human parainfluenza virus type 3 (HPIV3) morphogenesis, and α-gIs can block the first step of HPIV3 envelope glycoprotein processing, showing its antiviral ability [132]. Jasleen et al. further investigated the purified α-GI fraction and found that fraction AF1 exhibited antimicrobial activity against all tested human pathogens and significantly inhibited biofilm formation; Li et al. [133] isolated gallic acid (GA) and 3, 4, 6-tri-O-galloyl-D-glucose (TGG) from the fermentation products of tannic acid by *Aspergillus carbonarius* FCYN212, and the antioxidant and α-amylase/α-glucosidase inhibitory potential of TGG were also investigated; in addition, DNJ has been reported to have anticancer, anti-inflammatory, and anti-obesity potential [134,135,136]. Apparently, for α-gIs of microbial origin, a variety of other biological activities exist and need to be further investigated. In the future, these biological activities of α-gIs should be fully utilized and applied to the development of drugs and functional foods.

## 7. Conclusions and Prospects

In recent years, an increasing number of studies have shown that α-GI production by microorganisms has become a popular research direction, such as DNJ. However, the sources of reported α-GI-producing microbial strains are very limited, and the overall yield is not high and cannot reach the level of industrial application. Further efforts are needed to improve α-GI production through fermentation and metabolic strategies. On the other hand, the byproducts of industrial processing and industrial-grade food raw materials in fermentation are also a direction to explore. The raw materials used in food production, such as cereals, dairy products, fruits, and vegetables, contain various bioactive components, including flavonoids, polyphenols, and alkaloids. Fermentation with specific bacterial strains enhances the nutritional value of these raw materials and facilitates the development of hypoglycemic products enriched with diverse active ingredients. Moreover, conventional food processing often generates substantial amounts of byproducts that not only lead to resource wastage but also pose environmental challenges. However, employing bacterial strains for fermenting these byproducts into α-gIs offers an ingenious solution to this problem by valorizing inexpensive and noncompetitive carbon sources.

In addition, diabetes is a complex metabolic disease affected by many factors. To obtain better therapeutic effects, multidirectional and multitargeted therapy is essential, which requires not only a single drug formulation but also multiple drug formulations, making it important to find active substances that synergistically reduce blood glucose. There is often a synergistic relationship between compounds with hypoglycemic activity, and this aspect of plant-based α-gIs has been more intensively studied [137,138]. However, there are few studies on the specific interactions between multiple hypoglycemic substances in the fermentation broth. In future research, it is crucial to focus on the isolation and purification of these promising α-gIs. This will enable a comprehensive understanding of their mechanism of action and facilitate the practical application of this theory. In summary, this review will provide some guidance for microbial-derived α-gIs production, and we hope that more research results will flow into this field in the future, which will bring more possibilities for the prevention and treatment of diabetes.

## Figures and Tables

**Figure 1 foods-12-03344-f001:**
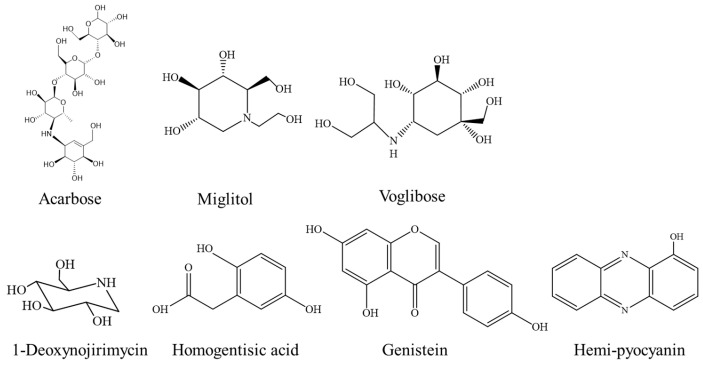
Structures of bacteria-based α-GIs possessing α-glucosidase inhibitory activity.

**Figure 3 foods-12-03344-f003:**
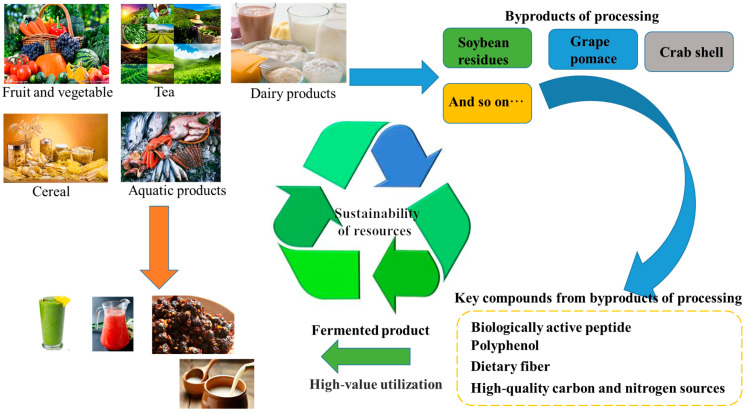
Development of fermented foods and reuse of byproducts from processing.

**Figure 4 foods-12-03344-f004:**
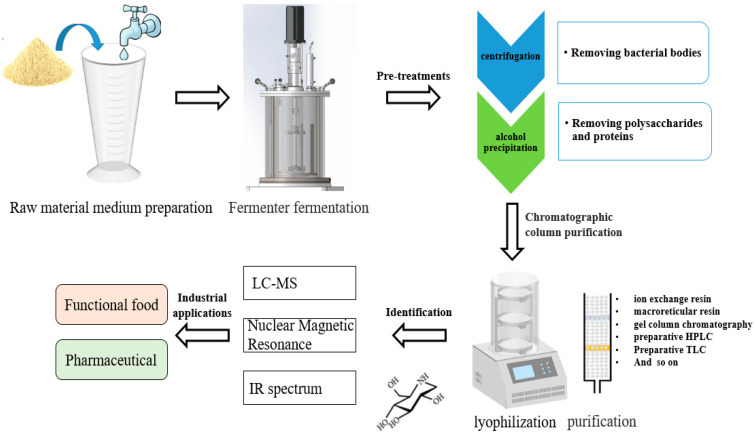
General procedures for novel α-Gis studies.

**Table 2 foods-12-03344-t002:** Current studies related to acarbose-producing microorganisms.

Strain	Carbon Source (g/L)	Nitrogen Source (g/L)	Culture Condition	Yield (mg/L)	Reference
*Actinoplanes* sp. CKD485-16	Maltose, 140 Glucose, 30	NZ-amine, 3	500 mOsm kg^−1^	3200	[60]
*Actinoplanes utahensis* ZJB-08196	Maltose, 50 Glucose, 30	Soybean meal	591 mOsm kg^−1^, 27 °C, 180 rpm, 7 days	4210	[61]
*Actinoplanes utahensis* ZJB-08196	Maltose, 14 Glucose, 6	Soybean meal, 9	27 °C, 200 rpm	4878	[62]
*Actinoplanes* sp. A56	Total sugar (75–80)	Peptone, 5	7.0–7.2 of pH, 40–50% of DO, 28 °C, 168 h	5000	[63]

**Table 3 foods-12-03344-t003:** Current studies related to DNJ-producing microorganisms.

Strain	Strategy	Yield (mg/L)	Reference
*B. amyloliquefaciens* DSM7	Lactose significantly increased 1-Deoxynojirimycin (DNJ) production at a C/N ratio of 6.25:1	1140	[66]
*B. methylotrophicus* K26	The maximum inhibitory activity was obtained under 4.61% sucrose, 7.03% yeast extract, and 34 °C culture condition	Not displayed	[67]
*B. subtilis* MORI	The optimal concentrations of galactose and soybean meal were 4.3% and 3.2%, respectively	824	[68]
*B. amyloliquefaciens* HZ-12	Gene overexpression and medium optimization, 20 g/kg lactose, and 10 g/kg malt extract were more favorable for DNJ production	1135.6	[69]
*Escherichia coli* as a heterologous host	Metabolic engineering	273	[70]
*B. subtilis* KCTC 13429	Random Mutagenesis and Culture Optimization were performed at 32 °C in 3.4% sorbitol and 2.4% yeast extract	773	[64]
*Streptomyces lawendulae*	The optimal conditions were obtained as follows: 11 days, 27 °C, pH 7.5, and 8% soluble starch content	42.875	[71]
*B. amyloliquefaciens* AS385	Sorbitol supplementation significantly increased DNJ production	460	[72]
*Streptomyces lavendulae*	Adding DNJ precursors, analogues and metabolic inhibitors increased the production	296.56	[65]

## Data Availability

Not applicable.

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
