# Peer review of "Research Progress of α-Glucosidase Inhibitors Produced by Microorganisms and Their Applications"

_foods, 2023, doi:10.3390/foods12183344_

Round 1

Reviewer 1 Report (Previous Reviewer 4)

The manuscript topic on Research progress of α-glucosidase inhibitors produced by microorganisms and their applications focusses on theoretical guidance for the development of microbial-derived hypoglycemic foods.

The topic chosen is novel and is very important from food biotechnology and development of innovative functional foods for diabetic patients.

The english language is fine & minor spell check is required.

Author Response

Dear Reviewer:

    Thank you for your acknowledgement and appreciation. I also appreciate your previous valuable feedback that helped enhance the manuscript’s quality.

Reviewer 2 Report (Previous Reviewer 3)

The manuscript has been improved, however some minor corrections are necessary

Line 102. Inhibits the activity of glucosidase not the hydrolysis. Glucosidase is not hydrolyzed, the carbohydrate is hydrolized.

Line 105. Delete “Therefore”

Line 139-140. Delete this sentence

Line 155. add and “some” mushrooms so not use italics for mushrooms

Line 182. delete this sentence

Lines 328-320, 333, 342 add italics to the species names

-in vitro and in vivo should be in italics

Line 361: replace lactobacillus by lactic acid bacteria

Line 367 add mali

Line 372: replace Lactobacillus plantarum by Lactiplantibacillus plantarum

Line 418: Dregea sinensis is a plant so, this sentence should be deleted or changed to another section.

Line 456. The section shouldn’t start with “in addition” please rephrase

Line 503. Please change the section title is meaningless

Line 510. change “,” by “.”

Line 514: lower case for “Day”

Line 583: replace “,” by “,.”before “In future”

Author Response

Dear Reviewer:

Thank you for your review. We have carefully revised the manuscript based on your suggestions, and have thoroughly examined the entire text, making additional revisions to address various detailed issues.

Please find the details of the revisions made in response to your suggestions attached.

Reviewer 3 Report (Previous Reviewer 2)

The authors have modified the manuscript as suggested which is now acceptable for publication.

Author Response

Dear Reviewer:

Thank you for your acknowledgement and appreciation. I also appreciate your previous valuable feedback that helped enhance the manuscript’s quality.

Reviewer 4 Report (Previous Reviewer 1)

This review paper can be used as a good scientific reference for further research related to microbial-based aGIs and the aGIs production from food-based materials. The quality of this manuscript was significantly increased in this submission. I suggested this manuscript may be considered for publication.

Author Response

Dear reviewer:

Thank you for your acknowledgement and appreciation. I also appreciate your previous valuable feedback that helped enhance the manuscript’s quality.

This manuscript is a resubmission of an earlier submission. The following is a list of the peer review reports and author responses from that submission.

Round 1

Reviewer 1 Report

This review paper can be used as a good scientific reference for further research related to microbial-based aGIs and the aGIs production from food-based materials. However, this manuscript should be significantly addressed some major points before further consideration:

- Please indicate clearly the novelty of this research compared to other review papers on the same topic and add to abstract and introduction part.

- Section 3.1 should be rewritten for more detail and clarity. The authors have confirmed that this work (Screening and evaluation of the potential strains) is very important.  However, very few related reports were mentioned and discussed in this part. Thus, should be updated much more research and discussion to clarify these issues.

- Line 189-193 should be removed.

- In Section 3.2, why only chose acarbose and DNJ for discussion? What about other inhibitors?

- Please recheck the scientific name of Probacitracin (PG) in reference 94. It's Prodigiosin.

- In Table 4, very few secondary metabolites were mentioned, almost in the form of crude extract or not mentioned. Thus, authors should classify clearly and state the name of the purified compound or crude extract. Few research conduct purification compounds which is also a notable point need perform in further reports in the future.

- Table 4 should be added a column for bioactivity (showing the IC50 value or inhibitory rate,…) and the column for presenting yield (It's necessary for confirmation of the aGIs production efficiency when using these fermented materials). Notably, in the conclusion part, the authors also confirm "the overall yield is not high" - lines 508-509, but no yield data was displayed.

- Overall, the authors should summarise what is the benefit of using food-related materials for the aGIs production. Furthermore, not only focusing on the interpretation of results, the authors also should indicate which raw materials are potential and promising for use in fermentation to produce aGI.

Author Response

Dear Reviewer,

Thank you for your constructive comments on my manuscript. We carefully considered your suggestions and made changes. Please find the specific response in the attachment.

Reviewer 2 Report

The review -Research progress of α-glucosidase inhibitors produced by micro-organisms and their applications-has attempted to provide comprehensive information on microbe derived αGIs.

However, I have some major concerns with the manuscript, which is why I am returning it promptly so that the authors can improve upon it and resubmit.

1.      English language needs to be greatly improved. I have tried to correct few pages but it is difficult to do so for the entire manuscript.

2.      The title says “and their applications’, but the authors have not included the applications of these microbe derived αGIs anywhere in the manuscript. Please add the applications as a separate heading.

3.      “Screening and evaluation of the potential strains”- should form a separate heading not under fermentation production, with different methods properly described.

4.      I feel that the review is a little unbalanced. Too much emphasis has been given to production by fermentation. This may be shortened and applications added.

5.      Other points

a)      Please give full form of an abbreviation where it first appears in the manuscript.

b)      Please number all subheadings serially. Please check-3.2.3, 3.2.4…

c)      Minor comments are marked on the attached pdf.

English language needs to be greatly improved. I have tried to correct few pages but it is difficult to do so for the entire manuscript

Author Response

Dear Reviewer

Thank you very much for your patience in revising and suggesting the manuscript, we have made serious revisions and refinements, please see the attached document

Reviewer 3 Report

This review describes the presence of different molecules with alpha glucosidase inhibitory activity from plants, fungi and obtained by metabolic activity of bacteria during growth in different substrates such as cereals, plants, dairy. There are lots of manuscripts and reviews about alpha glucosidase inhibitors. The English needs extensive editing. The manuscript is mainly a sequence of references without adding much interpretation of the bibliography used. The specific comments were added in the text attached (PDF file)

 The English needs extensive editing.

Author Response

Dear Reviewer:

First of all, thank you very much for your careful revision of the manuscript.in response to your suggestions, we have made serious revisions and refined the language.Please see annex for details

Reviewer 4 Report

The authors have presented a comprehensive overview of α-glucosidase inhibitors produced by microorganisms and their industrial applications. The topic chosen is quite pertinent and very little research information is available on the α-glucosidase inhibitors.

The english language is fine & only minor checks are required.

Author Response

Dear Reviewer :

We have touched up the manuscript and will upload it later! Thank you for supporting the manuscript!

Round 2

Reviewer 1 Report

Most of the suggestions were addressed.  The MS may be considered for publication.

Author Response

Dear Reviewer:

Your previous constructive suggestions on the manuscript are sincerely appreciated, while we have made further revisions based on the suggestions of other reviewers. Our final version will be uploaded later. 

Reviewer 2 Report

I am happy that the authors have made considerable improvement in the manuscript. However, language still needs to be improved further. Although the authors mention that” in fact, the third and fifth parts of our article are permeated with applications”, I still feel that “Applications“ should be a separate heading as you are attracting readers by that in your title. It should be easily available at one place and the reader should not have to extract it from different paragraphs.

English language still needs to be improved further

Author Response

Dear Reviewer:

Thank you very much for your advice. We have made the following specific changes:
Question: Although the authors mention that” in fact, the third and fifth parts of our article are permeated with applications”, I still feel that “Applications“ should be a separate heading as you are attracting readers by that in your title. It should be easily available at one place and the reader should not have to extract it from different paragraphs.

Response: Thank you very much for this suggestion, and we have revised the title of Part VI to: “Applications of microbial-derived α-GIs”, and added a paragraph: “Previously, we have presented the utilization of food-based raw materials and processing by-products for the production of α-glucosidase inhibitors (α-GIs) with applications in various fields such as hypoglycemic food development and drug discovery. In this section, we will discuss the extended biological activity of microbially derived α-GIs, providing valuable insights for future development and utilization of these compounds.”

Please see annex for additional questions.
